# Pre-Trained Language Models for Interactive Decision-Making

**Shuang Li [1*], Xavier Puig[1], Chris Paxton[2], Yilun Du[1], Clinton Wang[1], Linxi Fan[2],**
**Tao Chen[1], De-An Huang[2], Ekin Akyürek[1], Anima Anandkumar[2,3,†],**
**Jacob Andreas[1,†], Igor Mordatch[4,†], Antonio Torralba[1,†], Yuke Zhu[2,5,†]**

[1]MIT, [2]Nvidia, [3]Caltech, [4]Google Brain, [5]UT Austin
Junior authors are ordered based on contributions and senior authors[†] are ordered alphabetically.

## Abstract

Language model (LM) pre-training is useful in many language processing tasks. But can pre-trained LMs be further leveraged for more general machine learning problems? We propose an approach for using LMs to scaffold learning and generalization in general sequential decision-making problems. In this approach, goals and observations are represented as a sequence of embeddings, and a policy network initialized with a pre-trained LM predicts the next action. We demonstrate that this framework enables effective combinatorial generalization across different environments and supervisory modalities. We begin by assuming access to a set of expert demonstrations, and show that initializing policies with LMs and fine-tuning them via behavior cloning improves task completion rates by 43.6% in the Virtual-Home environment. Next, we integrate an active data gathering procedure in which agents iteratively interact with the environment, relabel past "failed" experiences with new goals, and update their policies in a self-supervised loop. Active data gathering further improves combinatorial generalization, outperforming the best baseline by 25.1%. Finally, we explain these results by investigating three possible factors underlying the effectiveness of the LM-based policy. We find that sequential input representations (vs. fixed-dimensional feature vectors) and LM-based weight initialization are both important for generalization. Surprisingly, however, the format of the policy inputs encoding (e.g. as a natural language string vs. an arbitrary sequential encoding) has little influence. Together, these results suggest that language modeling induces representations that are useful for modeling not just language, but also goals and plans; these representations can aid learning and generalization even outside of language processing. [2]

## 1   Introduction

**Language models** (LMs) play a key role in machine learning approaches to natural language processing tasks [9]. This includes tasks that are not purely linguistic, and require nontrivial planning and reasoning capabilities [24, 13]: for example, instruction following, vision-language navigation, and visual question answering. Indeed, some of these tasks are so distant from language modeling that one can ask whether pre-trained LMs can be used as a general framework even for tasks that involve no language at all. If so, how might these capabilities be accessed in a model trained only to process and generate natural language strings?

---

[*]Correspondence to: Shuang Li <lishuang@mit.edu>
[2]Project page: https://shuangli-project.github.io/Pre-Trained-Language-Models-for-Interactive-Decision-Making. Part of this work was done during Shuang's internship at NVIDIA.

36th Conference on Neural Information Processing Systems (NeurIPS 2022).

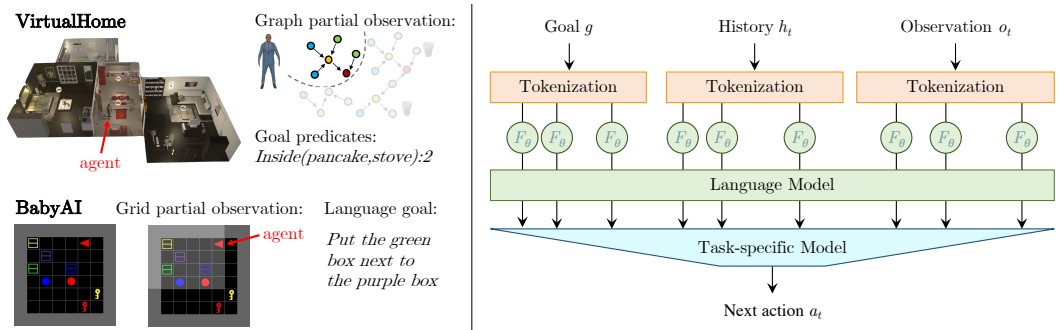

Figure 1: **Environments (left):** Different environments have different types of observations and goals. **Our approach (right):** We use pre-trained LMs as a general framework for interactive decision-making by converting policy inputs into sequential data. Such a method enables effective combinatorial generalization to novel tasks.

In this paper, we study these questions through the lens of **embodied decision-making**, investigating the effectiveness of LM pre-training as a general framework for learning policies across a variety of environments. We propose **LID**, a framework that uses Pre-Trained **L**anguage Models for **I**nteractive **D**ecision-Making. As shown in Figure 1 (right), we encode the inputs to a policy—including observations, goals, and history—as a sequence of embeddings. These embeddings are passed to a policy network initialized with the parameters of a pre-trained LM, which is fine-tuned to predict actions. This framework is broadly applicable, accommodating goals and environment states represented as natural language strings, image patches, or scene graphs.

We find that imitation learning using pre-trained LMs as policy initializers improves in-domain performance and enables strong generalization over novel tasks. For i.i.d. training and evaluation tasks, this approach yields 20% more successful policies than other baseline methods in Virtual-Home [31]. For combinatorial generalization to out-of-distribution tasks, *i.e.* tasks involving new combinations of goals, states or objects, LM pre-training confers even more benefits: it improves task completion rates by 43.6% for novel tasks (see Figure 3). These results hold for a variety of environment representations: encoding states as natural language strings, when possible, improves the data-efficiency of training, but even LMs fine-tuned on random environment encodings generalize combinatorially to new goals and states when trained on large enough datasets.

We further examine how our method may be used in environments where expert data is not available, and agents must instead actively gather data. To do this, we integrate an **A**ctive **D**ata **G**athering (**ADG**) procedure into pre-trained LMs as shown in Figure 2. Our proposed approach to ADG consists of three parts. First, exploration collects trajectories using a mix of random actions and actions generated by the current policy. Exploration is insufficient in this high dimensional problem and most of the trajectories will likely fail to achieve the end goal. A key insight is that even the failed trajectories contain useful sub-trajectories that solve certain sub-goals, and we relabel these goals in a hindsight relabeling stage. The relabeled goal describes what was achieved in the extracted sub-trajectory. The policy update stage samples relabeled trajectories to update the policy. The active data gathering procedure allows us to train the LM-policy without pre-collected expert data. It also outperforms reinforcement learning (RL) methods on embodied decision-making tasks and enables more effective generalization to novel tasks.

Finally, we investigate *why* LID contributes to generalization. We hypothesize three possible causes for the effectiveness of LM-based policy initialization: (1) the use of *language-based input encodings*, and more generally LMs' ability to reason about natural language strings; (2) the *sequential structure* of transformer inputs, in contrast to the fixed-sized observations used by most policy architectures, and (3) *task-general inductive bias* conferred by weight initialization with LM pretraining. We investigate (1) by encoding the policy inputs as different types of sequences. Different input encoding schemes have only a negligible impact on the performance: the effectiveness of language modeling is not limited to utilizing natural strings, but in fact extends to arbitrary sequential encodings. We study (2) by encoding observations with a single vector embedding, thereby removing its sequential structure. This operation significantly degrades the model's performance on novel tasks. Finally, we investigate (3) by learning the parameters of the policy from scratch. The success rate after removing the pre-trained LM weights drops by 11.2%, indicating that LM pretraining provides useful inductive bias for sequence processing even when sequences are not natural language strings.

To summarize, our work has four main contributions:

- First, we propose to use **pre-trained LMs as a general scaffold** for interactive decision-making across a variety of environments by converting all policy inputs into sequential data.

- Second, we demonstrate that **language modeling improves combinatorial generalization in policy learning**: initializing a policy with a pre-trained LM substantially improves out-of-distribution performance on novel tasks.

- Third, we integrate an **active data gathering** procedure into the proposed approach to further enable policy learning on environments without using pre-collected expert data.

- Finally, we perform several analyses to explain the generalization capabilities of pre-trained LMs, finding that natural strings are not needed to benefit from LM pre-training, but the sequential input encoding and weight pre-training are important.

These results point to the effectiveness of the proposed framework with pre-trained LMs as a general-purpose framework to promote structured generalization in interactive decision-making.

## 2  Related Work

In recent years, word and sentence representations from pre-trained LMs [29, 9, 33] have become ubiquitous in natural language processing [49, 30]. Some of the most successful applications of pre-training lie at the boundary of natural language processing and other domains, as in instruction following [13] and language-guided image retrieval [22].

**Learning representations of language.** From nearly the earliest days of the field, natural language processing researchers observed that representations of words derived from distributional statistics in large text corpora serve as useful features for downstream tasks [8, 11]. The earliest versions of these representation learning schemes focused on isolated word forms [25, 28]. However, recent years have seen a number of techniques for training (masked or autoregressive) language models to produce contextualized word representations (which incorporate information neighboring words in sentences and paragraphs) via a variety of masked-word prediction objectives [9, 47].

**Applications of pre-trained LMs.** LMs can be fine-tuned to perform language processing tasks other than language modeling by casting those tasks as word-prediction problems. Successful uses of representations from pre-trained models include syntactic parsing [19] and language-to-code translation [45]; successful adaptations of LM prediction heads include machine translation [49], sentiment classification [6] and style transfer [18]. A number of tasks integrate language and other modalities, including visual question answering and image captioning [48]. Recent works find that image representations can be injected directly into LMs' embedding layers [42].

**Policy learning and LM.** Traditional policy learning methods, such as PPO [37], DQN [27], DDPG [21], A3C [26], perform well on playing tasks on Atari, OpenAI gym [5], and MuJoCo [41]. Some of them might fail to solve more challenging tasks on embodied environments [31, 39]. Several recent papers [36, 17, 15] propose to use LM for policy learning. Frozen Pretrained Transformer (FPT) [23] demonstrates that pre-trained LMs require very little fine-tuning to *match* the performance of task-specific models on several image classification and numerical sequence processing tasks. Semi-Supervised Skill Learning with Latent Language (SL)[3] [38] shows that LMs can serve as an effective backbone for hierarchical policies that express plans as natural language strings [2, 4]. In this paper, we focus on building a general framework for decision-making tasks using pre-trained LMs, even when language is not provided as an input or output.

## 3  Decision-Making and Language Modeling

### 3.1  POMDPs and Policy Learning

We explore the application of LMs to general sequential decision-making tasks in partially observed environments. These tasks may be formalized as partially observable Markov decision processes (POMDPs). A POMDP is defined by a set of states, a set of observations, a set of actions, and a transition model $\mathcal{T}(s_{t+1}|s_t, a_t)$ that maps the current state and action to the next state. Importantly, in a POMDP setting, the observation $o_t$ only captures a portion of the underlying state $s_t$, and an

optimal decision-making strategy (a **policy**) must incorporate both the current observation and the history of previous observations and actions. In our experiments, policies are parametric models $\pi_\phi(a_t|g, h_t, o_t)$ that output the probability of an action given the goals $g$, history information $h_t = \{o_1, a_1, \cdots, o_{t-1}, a_{t-1}\}$, and partial observations $o_t$ of the current state $s_t$.

In Figure 1 (right), we show a high-level overview of the proposed method. We first convert all policy inputs into a sequence and provide them as input to a transformer encoder. Representations from this encoder model are then passed to a task-specific decoder that predicts actions. We collect a dataset of $N$ training trajectories $\mathcal{D} = \{d^i\}_{i=1}^N$, where each trajectory consists of a goal and a sequence of observations and actions: $d^i = \{g^i, o_1^i, a_1^i, \cdots, o_{T_i}^i, a_{T_i}^i\}$, where $T_i$ is the length of the trajectory. We then train the policy to maximize the probability of actions we want to achieve $\boldsymbol{a}^i = \{a_1^i, \ldots, a_{T_i}^i\}$ across trajectories using the cross-entropy loss:

$$\phi^* = \arg\min_\phi \left( - \sum_{i=1}^N \sum_{t=1}^{T_i} \ln \pi_\phi(a_t^i|g^i, h_t^i, o_t^i) \right). \tag{1}$$

## 3.2 Language models as policy initializers

Our experiments focus on **autoregressive**, **transformer-based LMs** [43]. These models are trained to fit a distribution over a text sequence $\boldsymbol{y} = \{y_i\}_{i=1}^n$ via the chain rule $p(\boldsymbol{y}) = p(y_1) \prod_{i=2}^n p(y_i \mid y_1, \ldots, y_{i-1})$. Each term on the right hand side is parameterized by a transformer network, which accepts the conditioned tokens as input. Each token passes through a learned embedding layer $F_\theta$, then the full conditioned sequence is fed into the LM. In our work, we use a standard LM, GPT-2, to process the input sequence rather than to predict future tokens.

Both POMDP decision-making and language modeling are naturally framed as sequence prediction tasks, where successive words or actions/observations are predicted based on a sequence of previous words or actions/observations. This suggests that pre-trained LMs can be used to initialize POMDP policies by fine-tuning them to model high-reward or expert trajectories, as described below.

# 4 Approach

We evaluate the effectiveness of pre-trained LMs in solving decision-making tasks across environments. We use **BabyAI** [16] and **VirtualHome** [31] to evaluate the proposed method. While both environments feature complex goals, the nature of these goals, as well as the state and action sequences that accomplish them, differ substantially across environments (Figure 1 (left)).

## 4.1 Policy Network

We first examine whether pre-trained LMs provide effective initializers when states and action histories are represented as natural language strings. We encode the inputs to the policy—including observations, goals, and action histories—as sequences of words. These word sequences are passed to the LM (using its pre-trained word embedding layer $F_\theta$) and used to obtain contextualized token representations. Token representations are averaged and used to predict actions. We design a policy network following the general policy framework proposed in Figure 1.

**Environment encodings in VirtualHome.** In VirtualHome, each goal consists of a sequence of predicates and multiplicities, and is translated into a templated English sentence (*e.g.* "Inside(apple, fridge):2" becomes "put two apples inside the fridge"). To encode the agent's partial observation, we extract a list of currently visible objects, their states (*e.g.* "open, clean"), and 3D world coordinates. We use a fully-connected layer to encode the 3D information and generate a feature representation of each object in the observation. To encode history, we store information about all previous actions and convert them into templated English sentences (*e.g.* "I have put the plate on the kitchen table and the apple inside the fridge").

**Environment encodings in BabyAI.** The observation by default is a $7 \times 7$ grid. We convert the observation into $7 \times 7$ text descriptions, *e.g.* "purple ball", "grey wall", "open door", and combine them into a long sentence. We then convert the history actions into text descriptions, *e.g.* "turn left" and "go forward". We combine the language instruction (without modification) with the observation and history text descriptions, and feed them to the pre-trained LM.

We note that the policy network described above does not strictly require that these encodings take the form of natural language strings—other encodings of the environment as a sequence also work (see Section 7). This framework could be also generalized to support pixel-based observations using discretization schemes like the one employed in the Vision Transformer [10].

**Action prediction.** We pool LM outputs into a "context representation" that is used to predict the next action. In training, we maximize the probabilities of demonstrated actions. In inference, we select the valid action with the highest probability. See **Appendix C.1** for details.

VirtualHome and BabyAI have quite different observation spaces, action spaces, and goal spaces; however, we show that embedding policy inputs as sequences and utilizing the pre-trained LM as a policy initializer, enables effective generalization to novel tasks on both environments. We note that LID is not limited to VirtualHome and BabyAI, but is straightforwardly applicable to other embodied environments, such as ALFRED [40] and iGibson [39].

### 4.2 Training

We first examine LID through imitation learning on data collected by experts in Section 4.2.1. We then show that integrating an active data gathering procedure into LID enables policy learning without using expert data in Section 4.2.2. We use VirtualHome as an example to explain the data gathering.

#### 4.2.1 Policy Learning with Expert Data

The policy model is first initialized from a pre-trained LM and then fine-tuned on data collected by experts. We build on the VirtualHome environment to collect a set of expert trajectories using regression planning [20] and create a **VirtualHome-Imitation Learning dataset**. Given a task described by goal predicates, the planner generates an action sequence to accomplish this task (See **Appendix E.1**). The planner has access to privileged information, such as information about the pre-conditions and effects of each action, allowing an agent to robustly perform tasks in partially observable environments and generate expert trajectories for training and evaluation.

#### 4.2.2 Policy Learning with Active Data Gathering

Collecting expert data is sometimes challenging. It may require privileged information of the environment or human annotations, which can be time-consuming and difficult to scale. A promising way to scale up supervision is Hindsight Experience Replay (HER) [3], which allows agents to learn from orders of magnitude more data without supervision. However, existing HER methods [12] focus on simple tasks with small state/action space and full observability. They cannot tackle more complicated embodied decision-making tasks, requiring nontrivial planning and reasoning or natural language understanding. LID with the active data gathering (**LID-ADG**) can be used in solving tasks in such environments.

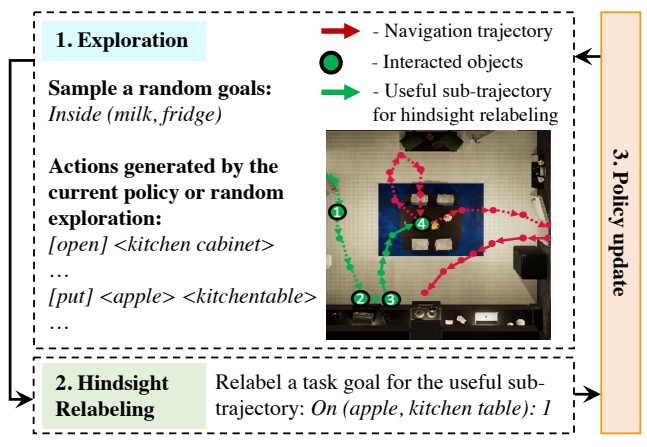

Figure 2: **LID with the active data gathering procedure.** By iteratively repeating the exploration, hindsight relabeling, and policy update, LID with active data gathering can learn an effective policy without using pre-collected expert data.

As shown in Figure 2, LID-ADG consists of three stages, *i.e.* **exploration**, **hindsight relabeling**, and **policy update**. The key idea is to gradually improve the task success rate by asking the agent to iteratively explore the environment, relabel failure samples, and update its policy using imitation learning. In the **exploration** stage, we first randomly sample a goal and an initial state. We then use a mix of random actions and actions generated by the current policy $\pi_\phi(a_t|g, h_t, o_t)$ to obtain the next action. We repeat this process until this episode ends. We collect $M$ trajectories and store them in the replay buffers. The generated actions in the early stages rarely complete the given task.

However, even the failed trajectories contain useful sub-trajectories that solve certain sub-goals. In the **hindsight relabeling** stage, we extract useful sub-trajectories and relabel a goal $g'$ for each of them. We design a goal relabel function $f_l$ that generates a goal based on the sequence of observations and actions using hand-designed templates. In practice, we implement the goal relabel function as a program (see **Appendix E.2**). The *hindsight relabeling* stage allows sample-efficient learning by reusing the failure cases. During **policy update**, the agent samples the data from the replay buffers and updates its policy network $\pi_\phi$.

By interleaving the exploration, hindsight relabeling, and policy update, LID-ADG can gradually improve the policy without requiring pre-collected expert data. In embodied environments with large action spaces, sparse rewards, and long-horizon planning, RL methods often struggle to obtain stable policy gradients during training. Our method enables sample-efficient learning from the sparse rewards by relabeling new goals for the bad samples that the agent fails to achieve. In addition, LID-ADG leverages the stability of supervised learning in the *policy update* stage, enabling it to outperform RL approaches on a wide range of decision-making tasks.

## 5  Experiment Setup

We evaluate the proposed method and baselines on VirtualHome and BabyAI.

### 5.1  VirtualHome

VirtualHome is a 3D embodied environment featuring partial observability, large action spaces, and long time horizons. We evaluate policies' performance from three aspects: (1) performance on in-distribution tasks; (2) generalization to novel scenes; and (3) generalization to novel tasks.

**In-Distribution.** The predicate types and their counts in the goal are randomly sampled from the same distribution as the training data. The objects are initially placed in the environment according to common-sense layouts (*e.g.* plates appear inside the kitchen cabinets rather than the bathtub).

**Novel Scenes.** The objects are placed in random positions in the initial environment without common-sense constraints (*e.g.* apples may appear inside the dishwasher).

**Novel Tasks.** The components of all goal predicates are never seen together during training (*e.g.* both plates and fridges appear in training goals, but Inside(plate, fridge) only appears in the test set. (See **Appendix F** for more details.)

We evaluate the success rates of different methods on each test set. A given episode is scored as successful if the policy completes its entire goal within the maximum allowed steps of the environment. On each of the 3 test subsets, we use 5 different random seeds and test 100 tasks under each seed. Thus there are 1500 examples used to evaluate each model.

### 5.2  BabyAI

BabyAI is a 2D grid world environment for instruction following. Observations in BabyAI are $7 \times 7 \times 3$ grids describing a partial and local egocentric view of the state of the environment. We evaluate the methods on four representative tasks: *GoToRedBall*, *GoToLocal*, *PickupLoc*, and *PutNextLocal*. Performing well on the test set requires the models to generalize to new environment layouts and goals, resulting in new combinations of tasks not seen in training. For each method, we compute success rates over 500 episodes on each task.

## 6  Experiments

We first show results of the proposed method and baselines for embodied decision-making tasks using expert data in Section 6.1. We then show our results when using actively gathered data in Section 6.2.

### 6.1  Embodied Decision Making with Pre-trained Language Model (LID)

#### 6.1.1  Results on VirtualHome

We evaluate the following methods:

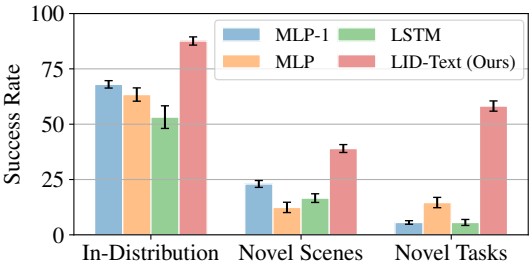

Figure 3: **Comparisons of the proposed method and baselines on VirtualHome.** All the methods are trained on expert data using imitation learning. *MLP-1*, *MLP*, and *LSTM* are baselines without using the pre-trained LM. The proposed method, LID-*Text (Ours)*, outperforms all baselines.

| Tasks | Methods | Number of Demos | | | | |
|---|---|---|---|---|---|---|
| | | 100 | 500 | 1K | 5K | 10K |
| **GoToRedBall** | BabyAI-Ori [16] | 81.0 | 96.0 | 99.0 | 99.5 | 99.9 |
| | LID-Text (Ours) | **93.9** | **99.4** | **99.7** | **100.0** | **100.0** |
| **GoToLocal** | BabyAI-Ori [16] | 55.9 | 84.3 | 98.6 | **99.9** | 99.8 |
| | LID-Text (Ours) | **64.6** | **97.9** | **99.0** | 99.5 | 99.5 |
| **PickupLoc** | BabyAI-Ori [16] | 28.0 | 58.0 | 93.3 | 97.9 | **99.8** |
| | LID-Text (Ours) | **28.7** | **73.4** | **99.0** | **99.6** | **99.8** |
| **PutNextLocal** | BabyAI-Ori [16] | **14.3** | 16.8 | 43.4 | 81.2 | 97.7 |
| | LID-Text (Ours) | 11.1 | **93.0** | **93.2** | **98.9** | **99.9** |

Table 1: **Success rates on BabyAI tasks.** All the methods are trained on offline expert data using imitation learning. LID-*Text (Ours)* outperforms BabyAI-Ori, the method used in the original paper [16].

**LID-Text (Ours)** is the proposed method that converts all environments inputs into text descriptions. The pre-trained LM is fine-tuned for decision-making (conditioned on goals, observations, and histories) as described in Section 4.1.

**Recurrent Network.** We compare our method with a recurrent baseline using an LSTM [14] to encode the history information. The hidden representation from the last timestep, together with the goal and current observation, are used to predict the next action.

**MLP** and **MLP-1**. We perform additional comparisons with baselines that do not use recurrent networks or pre-trained LMs. *MLP* and *MLP-1* take the goal, histories, and the current observation as input and send them to the multilayer perceptron neural network (MLP) to predict actions. *MLP-1* has three more average-pooling layers than *MLP* that average the features of tokens in the goal, history actions, and the current observation, respectively, before sending them to the MLP layer.

**Quantitative results.** Each method is trained on $20K$ demos from the VirtualHome-Imitation Learning dataset, and then evaluated on the three test subsets: **In-Distribution**, **Novel Scenes**, and **Novel Tasks**. In Figure 3, LID-*Text (Ours)*, which initializes the policy with a pre-trained LM, has higher success rates than other methods. This difference is most pronounced in the **Novel Tasks** setting, where test tasks require combinatorial generalization across goals that are never seen during training. Here, LID-*Text (Ours)* dramatically ($43.6\%$) improves upon all baselines. Such combinatorial generalization is necessary to construct general purpose agents, but is often difficult for existing approaches. Our results suggest that pre-trained LMs can serve as a computational backbone for combinatorial generalization.

### 6.1.2 Results on BabyAI

We use the standard training and test data provided by [16]. In BabyAI, performing well on unseen test tasks with new environment layouts and goals requires combinatorial reasoning. In Table 1, we report the success rate of models trained on different number of demos. **BabyAI-Ori** [16] is the method used in the original paper. **LID-Text (Ours)** is the proposed method that converts policy inputs into a text sequence. Given enough training data, *i.e.* 10K demos, both methods achieve high success rates, but LID-*Text (Ours)* outperforms BabyAI-Ori with less training data, indicating the proposed method improves sample efficiency when generalizing to novel tasks.

### 6.2 Pre-trained Language Model with Active Data Gathering (LID-ADG)

We compare **LID-ADG**, the proposed LM framework for decision-making using actively gathered data (Section 4.2.2), to a variety of baselines that do not use pre-collected expert data on VirtualHome.

**Random.** The agent selects the next action randomly from the valid action space at that state. **Goal-Object.** The agent randomly selects an object that in the goal and in the valid action space to interact with. For example, given a goal of "Inside(apple, fridge):1", this baseline might choose "grab apple", "open fridge", or other actions containing "apple" or "fridge". **Online RL.** We compare with PPO [37], one of the most commonly used online RL methods. For fair comparison, we equip PPO with the same main policy network as the proposed method. Our implementation is

| | In-Distribution | Novel Scenes | Novel Tasks |
|---|---|---|---|
| **Random** | $0.0 \pm 0.0$ | $0.0 \pm 0.0$ | $0.0 \pm 0.0$ |
| **Goal-Object** | $0.8 \pm 0.5$ | $0.0 \pm 0.0$ | $0.4 \pm 0.4$ |
| **PPO** | $0.0 \pm 0.0$ | $0.0 \pm 0.0$ | $0.0 \pm 0.0$ |
| **DQN+HER** | $0.0 \pm 0.0$ | $0.0 \pm 0.0$ | $0.0 \pm 0.0$ |
| **LID-ADG (Ours)** | $\mathbf{46.7 \pm 2.7}$ | $\mathbf{32.2 \pm 3.3}$ | $\mathbf{25.5 \pm 4.1}$ |

Table 2: **Comparisons of methods without using expert data on VirtualHome.** LID-*ADG (Ours)* is the only successful approach.

| | In-Distribution | Novel Scenes | Novel Tasks |
|---|---|---|---|
| **LID-ADG (Ours)** | $46.7 \pm 2.7$ | $\mathbf{32.2 \pm 3.3}$ | $25.5 \pm 4.1$ |
| **PPO (LID-ADG Init)** | $\mathbf{53.7 \pm 3.5}$ | $30.2 \pm 3.4$ | $\mathbf{27.8 \pm 2.7}$ |
| **DT (LID-ADG Data)** | $42.4 \pm 1.5$ | $21.6 \pm 2.48$ | $16.8 \pm 1.0$ |

Table 3: The proposed method with active data gathering, LID-ADG (Ours), can be used as an policy initializer for online RL or a data provider for offline RL.

based on Stable Baselines3 [35]. **Hindsight Experience Replay.** We compare with DQN+HER used in [3] and modify its main policy network to be the same as the proposed method.

**Quantitative results**. We compare LID-ADG with baselines on VirtualHome in Table 2. Each experiment is performed 5 times with different random seeds. The **Random** baseline is always 0, indicating the tasks in VirtualHome cannot be easily solved by a random policy. **Goal-Object** is better than *Random* because *Goal-Object* has access to objects in the goal and it samples actions from a much smaller action space. The online RL baseline, **PPO**, fails to solve tasks in VirtualHome featured by partially observation, large state/action space, and long-term horizon. **DQN+HER** works well on simple tasks on 2D environments, but they cannot tackle VirtualHome tasks neither, requiring nontrivial planning and reasoning. LID-ADG does not require expert data and can solve the complicated tasks in 3D embodied environments which cannot be easily achieved using RL. [3]

**Policy initializer and data provider.** LID-ADG can further be used to initialize the weights for fine-tuning RL policies and to gather data for offline learning. As shown in Table 2, directly training RL, *e.g.* PPO, fails to solve tasks in VirtualHome. However, after using the policy trained by LID-ADG to initialize the PPO policy, we may effectively learn an interactive policy with good performance. In Table 3, **PPO (LID-ADG Init)** is initialized from LID-ADG and further fine-tuned to solve the tasks in VirtualHome. After initialization, PPO improves its success rate by $53.7\%$ on the *In-Distribution* setting (See PPO results in Table 2 and Table 3). In addition, LID-ADG can provide data for offline learning. LID-ADG saves the relabeled data in replay buffers. We train Decision Transformer (DT) [7] using the data collected by LID-ADG. See **DT (LID-ADG Data)** in Table 3.

# 7 Analysis: Understanding the Sources of Generalization

The pre-trained LM policy, fine-tuned on either expert data or actively gathered data, exhibits effective combinatorial generalization. Is this simply because LMs are effective models of relations between natural language descriptions of states and actions [1], or because they provide a more general framework for combinatorial generalization in decision-making? We hypothesize and investigate three possible factors to understand the sources of such combinatorial generalization. We use policies trained on the expert data as an example to explain the experiments.

## 7.1 Input Encoding Scheme

We first hypothesize that converting environment inputs into natural language contributes to the combinatorial generalization as the LMs are trained on language data. We explore the role of *natural language* by investigating three alternative ways of encoding policy inputs to our model without using natural language strings: two in VirtualHome, and one in BabyAI. BabyAI results are in Appendix A.

**Index encoding in VirtualHome.** Rather than natural language strings, LID-*Index (Ours)* converts policy inputs into integer indices. LID-*Index (Ours)* retains the discrete, serial format of the goal, history, and observation, but replaces each word with an integer, and replaces the embedding layer from the pre-trained LM with a new embedding layer trained from scratch. For example, *grab apple* is mapped to (5,3) based on the positions of *grab* and *apple* in the vocabulary set.

**Unnatural string encoding in VirtualHome.** LID-*Unnatural (Ours)* replaces the *natural language* tokens (e.g. converting the goal "On(fork, table):1" to *put one fork on the table*) with random ones (e.g. converting On(fork, table) to *brought wise character trees fine yet*). This is done by

---

[3]Note that the results of LID-Text in Figure 3 and results of LID-ADG in Table 2 are not directly comparable because the difficulty level of the evaluated tasks are different. See **Appendix F** for more details.

Table 4: **Success rates of policies trained with different input encodings in the *Novel Tasks* setting on VirtualHome.** The text encoding is the most sample-efficient, but all models converge to similar performance given sufficient training data.

| Methods | Number of Demos | | | | | |
|---|---|---|---|---|---|---|
| | **100** | **500** | **1K** | **5K** | **10K** | **20K** |
| **LID-Text (Ours)** | $\mathbf{8.8 \pm 1.4}$ | $\mathbf{22.2 \pm 1.7}$ | $26.8 \pm 1.0$ | $46.0 \pm 1.0$ | $\mathbf{58.2 \pm 1.2}$ | $58.2 \pm 1.6$ |
| **LID-Index (Ours)** | $6.4 \pm 0.6$ | $18.0 \pm 3.8$ | $18.8 \pm 1.0$ | $45.5 \pm 2.1$ | $54.6 \pm 0.8$ | $57.8 \pm 0.9$ |
| **LID-Unnatural (Ours)** | $6.8 \pm 1.3$ | $18.6 \pm 2.1$ | $\mathbf{27.0 \pm 1.1}$ | $\mathbf{47.2 \pm 1.7}$ | $55.8 \pm 0.8$ | $\mathbf{58.8 \pm 0.9}$ |

randomly permuting the entire vocabulary, mapping each token to a new token. Such a permutation breaks the semantic information in natural strings.

LID-*Index (Ours)* and LID-*Unnatural (Ours)* have the same policy network as LID-*Text (Ours)*. All are fine-tuned on the expert data. The averaged results using 5 different random seeds on the Novel Tasks setting are reported in Table 4. Given few training data, e.g. 100 demos, all the models perform poorly, with success rates lower than $10\%$. LID-*Text (Ours)* achieves higher success rates than LID-*Index (Ours)* and LID-*Unnatural (Ours)* when dataset size increases, *e.g.* LID-*Text (Ours)* is around $4\%$ higher than LID-*Index (Ours)* and LID-*Unnatural (Ours)* with 500 training demos. When the training dataset is further enlarged, *e.g.* 20K demos, success rates of all approaches reach similar performance. This result indicates that the effectiveness of pre-trained LMs in compositional generalization is not unique to natural language strings, but can be leveraged from arbitrary encodings, although adapting the model to arbitrary encodings may require more training data.

## 7.2 Sequential Input Representation

Next, we explore whether generalization requires the sequential processing mechanisms in transformer-based LMs. We investigate whether the LM pre-trained policy will still be effective when the input encoding is not sequential. **No-Seq** encodes the goal as a single vector by averaging all goal embeddings. History and observation features are obtained in the same way. All features are then sent to the pre-trained LM to predict actions. As shown in Table 5, removing sequential structure significantly hurts performance on *Novel Tasks*. No-

Table 5: **Experiments on sequential inputs and weight initialization.** Fine-tuning the pre-trained weights and the usage of sequential encoding are important for combinatorial generalization.

| | In-Distribution | Novel Tasks |
|---|---|---|
| **LID-Text (Ours)** | $87.6 \pm 1.9$ | $\mathbf{58.2 \pm 2.3}$ |
| **No-Seq** | $74.0 \pm 2.3$ | $2.0 \pm 0.6$ |
| **No-Pretrain** | $\mathbf{90.8 \pm 2.0}$ | $47.0 \pm 2.8$ |
| **No-FT** | $51.2 \pm 4.5$ | $17.0 \pm 2.9$ |

*Seq* achieves good performance on test tasks that are closer to training tasks, but cannot generalize well to more challenging unseen tasks. Thus, combinatorial generalization in pre-trained LMs may be attributed in part to transformers' ability to process sequential input representations effectively.

## 7.3 Favorable Weight Initialization

Finally, we investigate if the favorable weight initialization from LM pre-training enables effective generalization of the proposed model. **No-Pretrain** does not initialize the policy using the pre-trained LM, but instead training the policy on the expert data from scratch. In Table 5, we find that removing the pre-trained weights can fit the in-domain data and thus performs well on the *In-Distribution* setting. However, its success rate is $11.2\%$ lower than the proposed model on the *Novel Tasks* setting, indicating the pre-trained weights are important for effective generalization, but not necessary for effective data fitting. We further test a baseline, **No-FT**, that keeps the pre-trained weights of the language model but freezes them while training the rest model on our expert data. Freezing the pre-trained weights without fine-tuning significantly hurts the performance on both settings, suggesting that fine-tuning of the transformer weights is essential for effective combinatorial generalization.

Together, these results suggest that sequential input representations (vs. fixed-dimensional feature vectors) and favorable weight initialization are both important for generalization, however, the input encoding schemes (e.g. as a natural language string vs. an arbitrary encoding scheme) has little influence. These results point to the potential broader applicability of pre-trained LMs as a computational backbone for compositional embodied decision making, where arbitrary inputs, such as language, images, or grids, may be converted to sequential encodings.

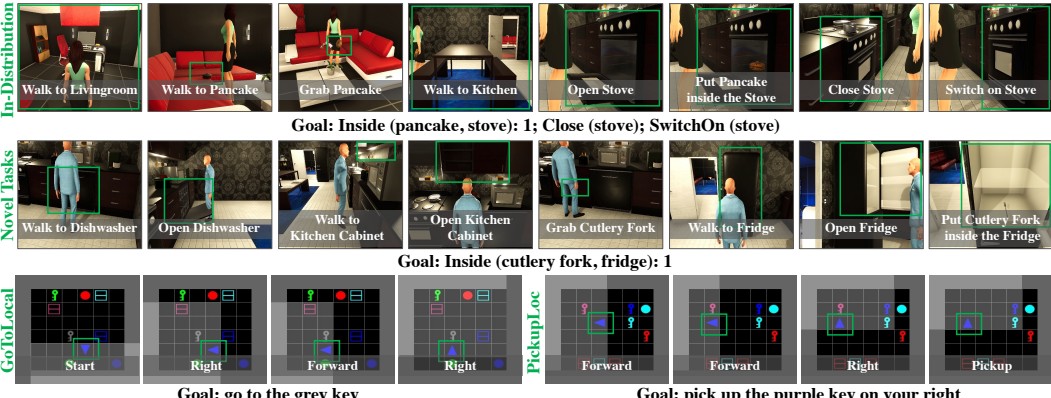

Figure 4: **Qualitative results of our model on VirtualHome and BabyAI.** We only show a sub-trajectory in each example to save space. The interacted objects are labelled by green bounding boxes.

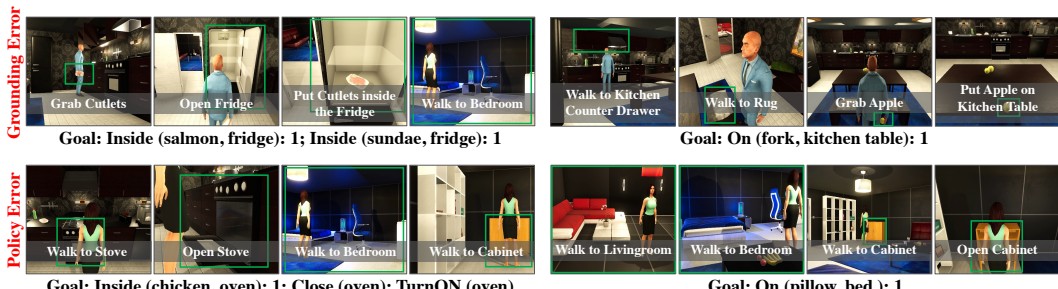

Figure 5: **Failure cases.** We show failure cases caused by the grounding error and policy error. The interacted objects are labelled by green bounding boxes.

## 8 Qualitative Results

In Figure 4, we show examples of LID-*Text (Ours)* completing tasks in VirtualHome and BabyAI. We show two successful examples from VirtualHome on the *In-Distribution* and *Novel Tasks* settings, and two successful examples from BabyAI on solving the *GoToLocal* and *PickupLoc* tasks. We only show short trajectories or extract a sub-trajectory for saving space.

**Failure case analysis.** In Figure 5, we show some failure cases of the proposed method. We observed two main types of failure cases: grounding error and policy error. For failures caused by the grounding error, the agent interacts with a wrong object that is not related to the given goal, *e.g.* the agent puts *cutlets* instead of the *salmon* inside the fridge. For failures caused by the policy error, the agent cannot find the target objects or does not interact with them. The proposed method that converts policy inputs into sequential encodings and feeds them to the general LM framework can accomplish decision-making tasks efficiently, however, there are still challenging tasks that the policy fails to accomplish. Larger LMs, *e.g.* GPT-3 [6], may improve the success rate of those challenging tasks.

## 9 Conclusion and Broader Impact

In this paper, we introduced LID, a general approach to sequential decision-making that converts goals, histories, and observations into sequences and processes them using a policy initialized with a pre-trained LM. We integrated an active data gathering procedure into the proposed method to enable policy learning without using expert data. Our analysis showed that input representation and favorable weight initialization both contribute to the generalization while the input encoding scheme has little influence. One drawback of the active data gathering is that it relies on hand-designed rules for task relabeling. More generally, a potential disadvantage of the proposed approach is that biases of the pre-trained LMs may influence its behavior, and further study of LID-based models' bias is required before they may be deployed in sensitive downstream applications. Nevertheless, our results demonstrate that LID enables effective combinatorial generalization across different environments, and highlight the promise of LM pre-training for more general decision-making problems.

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
