# OpenReview forum: "Pre-Trained Language Models for Interactive Decision-Making"
_NeurIPS.cc/2022/Conference — NeurIPS 2022 Accept_

### Official Review · Reviewer_EQi5 · 2022-07-11

**Rating:** 7
**Confidence:** 3
**Soundness:** 3 good
**Presentation:** 3 good
**Contribution:** 4 excellent

**Summary:**

This paper introduces LID, a framework that encodes the goal, history, and observation as the tokens and fine-tunes a pre-trained GPT-2 on the next action prediction task. The authors show empirically that such a pre-training mechanism significantly improves performance on both VirtualHome and BabyAI tasks. The proposed method is especially useful when only a limited number of demos are available. Their LID-ADG is able to make meaningful predictions on VirtualHome with any expert data (unlike baselines which get about 0% accuracy.


**Questions:**

- Are all the tokens in brackets (such as [grab] and <apple>)  treated as individual special tokens and added to the tokenizer or they are just treated as normal takes?
- I don't see the reason why the authors only use autoregressive language model. The next action prediction task doesn't seem to require an autoregressive model since it's not a sequence prediction task. I wonder how bidirectional Transformers such as BERT and RoBERTa perform.


**Limitations:**

Yes, the authors discuss them in Section 9.

**Strengths And Weaknesses:**

### Strengths
- The proposed method is novel, simple, and quite effective.
- The authors conduct a deep analysis in Section 7 to provide deeper understanding of various design choices in LID.
- The paper is easy to read.

### Weaknesses
- One concern is about the reproducibility of the paper. The authors do not provide the detailed architecture and hyperparameters of the method. Although there is a short description in Appendix D.2. I don't think it is sufficient for reproduction.
- It would be better if the authors can provide standard deviations in Tables 1 and 4. With a small number of demos, I believe that the standard deviation would be large.

### Summary
Overall, the proposed method is sound and effective, so I recommend accepting it. However, the tasks studied in this paper are not in my research area, so I am not confident with my judgment.

### Violating the formatting instruction
Notably, the authors include the appendix at the end of the main paper submission instead of putting it in the supplementary material which violates the NeurIPS format. I wonder if it is acceptable.

---

> ### Author Response · Authors · 2022-08-02
> **Response to Reviewer EQi5**
>
> Dear Reviewer, thank you for your valuable comments and feedback. We appreciate that you think our paper is novel, simple, and quite effective. We have addressed your questions below. Please let us know if you have any additional questions -- we are happy to clarify or provide additional experiments.
>
> ------------------------------------
>
> **Q1: One concern is about the reproducibility of the paper. The authors do not provide the detailed architecture and hyperparameters of the method. Although there is a short description in Appendix D.2. I don't think it is sufficient for reproduction.**
>
> **A:** Thanks a lot for your suggestion. In the paper we submitted, we provided the detailed model architecture in Appendix D.1, Figure 6, and Figure 7. The training details and hyperparameters are provided in Appendix D.2.
>
> In this rebuttal, we further added our code for training LID-Text in the supplementary materials.
>
> ------------------------------------------
>
> **Q2: It would be better if the authors can provide standard deviations in Tables 1 and 4. With a small number of demos, I believe that the standard deviation would be large.**
>
> **A:** Thanks for your suggestion. We have added the standard deviations in Table 1 and Table 4 in the updated version.
>
> ------------------------------------------
>
> **Q3: Violating the formatting instruction. Notably, the authors include the appendix at the end of the main paper submission instead of putting it in the supplementary material which violates the NeurIPS format.**
>
> **A:** Dear reviewer, adding the appendix at the end of the main paper does not violate the NeurIPS format.
>
> On the NeurIPS webpage (https://neurips.cc/Conferences/2022/PaperInformation/StyleFiles), they have explicitly mentioned that “additional pages containing only the checklist, references, and appendices are allowed.”
>
> Also on the NeurIPS FQA webpage (https://neurips.cc/Conferences/2022/PaperInformation/NeurIPS-FAQ), they have answered this question “Yes. You can include appendices with the main submission file, or you can include them as a separate file in the supplementary materials.”
>
> The provided NeurIPS template also contains the appendix at the end (page 6 in https://media.neurips.cc/Conferences/NeurIPS2022/Styles/neurips_2022.pdf).
>
> -------------------------------------
>
> **Q4: Are all the tokens in brackets (such as [grab] and <apple>) treated as individual special tokens and added to the tokenizer or they are just treated as normal takes?**
>
> **A:** They are treated as normal tokens. [grab] <apple> is first converted to “grab apple” and then sent to the tokenizer.
>
> ---------------------------------------
>
> **Q5: I don't see the reason why the authors only use autoregressive language model. The next action prediction task doesn't seem to require an autoregressive model since it's not a sequence prediction task. I wonder how bidirectional Transformers such as BERT and RoBERTa perform.**
>
> **A:** We agree that the autoregressive language model is not the only way to do action prediction. GPT-2 is one of the most representative language models, and thus we opted to use GPT-2 in our method. However, other language models such as BART [15], BERT, and RoBERTa can also be used in our framework.
>
> We have added an experiment that replaced the GPT-2 with BART, which has a Bidirectional Encoder similar to BERT. See Fig 1 in [15] for the comparisons of BART, BERT, and GPT. In our experiment, we used the pre-trained BART model (BART-BASE) from the HuggingFace library (https://huggingface.co/docs/transformers/model_doc/bart).
>
> We compared the results of using GPT-2 and BART in the LID-ADG framework. The results of using BART on the three test settings, In-Distribution, Novel Scenes, and Novel Tasks, are 49.0 (std. 3.7), 38.0 (std. 6.1), and 33.7 (std. 2.6), respectively. The result of using GPT-2 (results reported in Table 2) on these three test settings are 46.7 (std. 2.7), 32.2 (std. 3.3), and 25.5 (std. 4.1), respectively. The bidirectional transformer (BART) works well on solving decision-making tasks as well.
>
> [15] BART: Denoising Sequence-to-Sequence Pre-training for Natural Language Generation, Translation, and Comprehension

---

> > ### Comment · Reviewer_EQi5 · 2022-08-08
> > **Response to Authors' feedback**
> >
> > I have read the authors' feedback and the updated version of the paper. I appreciate that the authors add additional experimental results on using bidirectional encoders from BART and show that it works even better. My main concern regarding reproducibility is also resolved by the provided code. However, I still believe the description in Appendix D.2 provides enough training details. The batch size, number of training steps, weight decay of AdamW, and learning rate scheduler are all missing. Although practitioners can look them up from the code, I would still recommend including sufficient details in the paper.
> > Overall, I would like to keep my original rating and recommend accepting this paper.

---

### Official Review · Reviewer_uuzZ · 2022-07-11

**Rating:** 7
**Confidence:** 4
**Soundness:** 4 excellent
**Presentation:** 3 good
**Contribution:** 3 good

**Summary:**

This paper proposes to utilize the pre-trained language model (PLM) for solving interactive decision-making problem, named LID. Specifically, the policy network consists of a PLM and tailored task modules. The policy information (goal, history, information) will be described as the natural text and encoded by GPT-2 PLM. During the fine-tuning process, the PLM and task-specific modules will be jointly optimized (the authors also study the different training paradigms). To alleviate the scarcity problem of expert data, the authors also propose an active data gathering method to make the agent learn in a self-supervised way. The BabyAI platform and VirtualHome platform are used to evaluate the performance of the LID framework. Several empirical experiments and the corresponding analysis show the rationality and generalization ability of the proposed LID framework.

The contributions of this paper are two-folded: 1) proposing a PLM-based framework for interactive decision-making task, i.e., LID in this paper; 2) proposing a self-supervised data gathering method, which provides a stable RL training process. The authors also analyze the source of the generalization ability of LID framework, which can provide a clear direction for future research.


**Questions:**

1. Pre-trained language models limit the length of input text. For instance, the maximum input length of GPT-2 [1] is 1024 [2]. Will the length of encoded text in your experiments exceed this number? If not, can you figure out a solution for handling long-text input? Because the environment and policy description could be too long, we may not neglect this situation.
2. In the exploration stage (Section 4.2.2), what is the exact sampling method for the goal and initial state?
3. The BabyAI 1.1 baselines [3] report the mean ± std in their papers. However, the evaluation of the BabyAI in this paper (Table 1 and 4) did not use different random runs. I did not find any justification in Section 5.2. Could you please offer an explanation?

[Rebuttal Updates] I confirmed the authors' answers.

Reference:

[1] Radford, A., Wu, J., Child, R., Luan, D., Amodei, D., & Sutskever, I. (2019). [Language models are unsupervised multitask learners.](https://d4mucfpksywv.cloudfront.net/better-language-models/language-models.pdf).

[2] https://huggingface.co/gpt2/blob/main/config.json

[3]  Hui, D. Y. T., Chevalier-Boisvert, M., Bahdanau, D., & Bengio, Y. (2020). [BabyAI 1.1.](https://arxiv.org/abs/2007.12770)

**Limitations:**

- Societal Impact: No potential negative societal impact. The authors provide a new perspective to aid policy learning with a pre-trained language model.
- Limitation: 1) Building text descriptions for each task still requires human labor. We do not know what textual format is optimal for policy learning. It varies from task to task, model to model. On the other hand, as I stated in Question 1, the long-text input could restrict the scalability of this framework. 2) The proposed methods also need humans to design some templates/rules, as the authors mentioned in the conclusion part.


**Strengths And Weaknesses:**

**Strengths**:
1. ***Novelty***: It is not surprising that the "pre-training LM then fine-tuning" paradigm is widely used in natural language processing tasks. Although the authors also follow this paradigm, it is interesting to see that the sequence modeling ability of PLM can be used in policy learning.
2. ***Significance***: The proposed framework is well supported by empirical studies and evaluated on two commonly used benchmarks. This new framework powered by PLM can be beneficial to the research community of policy learning.
3. ***Quality***: This paper is technically sound, and the authors provide enough technical details in the Appendix including network architecture, encoding methods, etc.

**Weaknesses**:
1. ***Scalability***: The proposed encoding method is templated-based (Line 155-156). Although the input encoding scheme (Section 7.1) may be a trivial problem, the encoding scheme may still affect the performance. Searching for the optimal encoding scheme is an expensive process, which may bring a high cost of hand-crafted engineering.  Besides, the data gathering method also relies on hand-designed templates (Line 220).
2. ***Presentation***: The related work of PLM is adequately cited. But the authors should also introduce the background of policy learning so that the significance of this work can be highlighted.
3. ***Performance***: Compared to the work that uses traditional networks like DQN, the integration of PLM may affect the inference speed.
4. ***Clarity***: Most parts of this paper are well written. However, there are some typos in the paper:
    - Line 53: pretrained LMs -> pre-trained LMs
    - Line 104: language -> language. (missing full stop mark)
    - Some papers should be cited in a proper way: Line 108: [23], Line 109: [36], Line 285:[15], Line 287 [15]. For example, in Line 108, "[23] show that" needs to be rewritten as "Frozen Pretrained Transformer (FPT) [23] show that".

[Rebuttal Updates] The authors provided the additional experiments for addressing my concern of scalability. The authors also revised the typos and added the related works.

---

> ### Author Response · Authors · 2022-08-02
> **Response to Reviewer uuzZ (1/2)**
>
> Dear Reviewer, thank you for your valuable comments and feedback. We appreciate that you think our paper is novel, technically sound, and has significant contributions. We have addressed your questions below. Please let us know if you have any additional questions -- we are happy to clarify or provide additional experiments.
>
> ----------------------------------------
>
> **Q1: Scalability: The proposed encoding method is templated-based (Line 155-156). Although the input encoding scheme (Section 7.1) may be a trivial problem, the encoding scheme may still affect the performance. Searching for the optimal encoding scheme is an expensive process, which may bring a high cost of hand-crafted engineering. Besides, the data gathering method also relies on hand-designed templates (Line 220).**
>
> **A:** Thanks a lot for your suggestion. We agree that the encoding scheme may affect the performance. However, the influence of using different templates is small after fine-tuning the model on enough data, as shown in Table 4. To further demonstrate the scalability of the proposed method, we add an experiment where the model (LID-ADG) is trained on templated English but is used to solve natural language tasks written by humans during testing (the Real-Human-Goal setting).
>
> We collected 16,114 language goals by combining the collected human language descriptions and objects from VirtualHome. These language descriptions are different from the templated English used during training. We found that our model generalizes from the rigid English templates used in training and can understand a diverse set of naturally written English goals at test time.
>
> We tested our model 5 times using different random seeds. At each time, we randomly select 1000 examples. The performance of LID-ADG on this Real-Human-Goal setting is 41.2%. Its performance in the In-Distribution setting is 46.7%, as shown in Table 2. The difficulty of tasks generated by humans is close to that of tasks in the In-Distribution setting, but the language descriptions made by humans are much more diverse. The results demonstrate that our method trained on templated English has the scalability to solve more general human tasks. (The human experiments were approved by the institutional review board (IRB). Please see Appendix I for more details about this human experiment.)
>
> -------------------------------------------
>
> **Q2: Presentation: The related work of PLM is adequately cited. But the authors should also introduce the background of policy learning so that the significance of this work can be highlighted.**
>
> **A:** Thanks a lot for your suggestion. We have added the policy learning literature in the related work section in the update paper.
>
> -------------------------------------------
>
> **Q3: Clarity: Most parts of this paper are well written. However, there are some typos in the paper.**
>
> **A:** Thanks a lot for your suggestion. We have corrected all the mentioned typos in the updated paper.

---

> > ### Author Response · Authors · 2022-08-02
> > **Response to Reviewer uuzZ (2/2)**
> >
> > **Q4: Pre-trained language models limit the length of input text. For instance, the maximum input length of GPT-2 [1] is 1024 [2]. Will the length of encoded text in your experiments exceed this number? If not, can you figure out a solution for handling long-text input? Because the environment and policy description could be too long, we may not neglect this situation.**
> >
> > A: This is a great question. In our experiments, the length of the encoded text is smaller than the default input length (1024) of GPT-2. However, in general, we agree that the model should be able to handle inputs of arbitrary length.
> >
> > There exists a large body of existing work on supporting long-length inputs in transformers. Some works shorten sequences that exceed the max input length by summarizing contextual information [5]. We used a similar approach to represent each object node in the observation. Instead of using a long sentence to describe the object’s name, state, and spatial relations, we use a single feature vector to describe each node (see Appendix D and Fig 7).
> >
> > Other works split a long sequence into chunks and use a task-specific model to aggregate the outputs of the chunks [6,7].
> >
> > Since the input sequence length of Transformers is limited in part by the quadratic time and memory complexity of attention, many works have further developed more scalable parameterizations of attention that readily extend to longer sequences. Sparse attention is used in Big Bird [8] and sparse transformers [9], and a low-rank factorization is used in Linformer [10]. Longformer [11] combines local windowed attention with sparse global attention. Performers [12] and Linear Transformers [13] develop kernel-based approaches.
> >
> > Handling long-text input continues to be a rich and active research topic, and such techniques are complementary to the framework and methods we proposed and may be directly applied. We hope to explore these strategies in policy learning applications in future work.
> >
> > [5] Recursively Summarizing Books with Human Feedback
> >
> > [6] Hierarchical Transformers for Long Document Classification
> >
> > [7] Recurrent Chunking Mechanisms for Long-Text Machine Reading Comprehension
> >
> > [8] Big Bird: Transformers for Longer Sequences
> >
> > [9] Generating long sequences with sparse transformers.
> >
> > [10] Linformer: Self-attention with linear complexity
> >
> > [11] Longformer: The Long-Document Transformer
> >
> > [12] Rethinking Attention With Performers
> >
> > [13] Transformers are RNNs: Fast autoregressive transformers with linear attention.
> >
> > --------------------------------------
> >
> > **Q5: In the exploration stage (Section 4.2.2), what is the exact sampling method for the goal and initial state?**
> >
> > **A:** As shown in Appendix Algorithm 2, we first generate a set of initial states in VirtualHome using the code released by [14]. For each initial state, we are able to get a set of feasible tasks that can be accomplished in this environment. For example, in an initial state, if the apple is on the kitchen table, a feasible task goal could be “put the apple inside the fridge.” In contrast, “put the banana inside the fridge” is not a feasible task if there is no banana in the initial state.
> >
> > We collected 9,893 initial states and randomly sampled an initial state and its feasible goal every time when we reset the environment. After each data collection interaction, we obtain a set of new goals using the goal relabel function. We save the goal and its corresponding initial state in the replay buffers and use the same strategy to sample the goal and initial state in the next interaction. We have added these details in the updated paper in Appendix F.2.
> >
> > [14] Watch-And-Help: A Challenge for Social Perception and Human-AI Collaboration
> >
> > --------------------------------
> >
> > **Q6: The BabyAI 1.1 baselines [3] report the mean ± std in their papers. However, the evaluation of the BabyAI in this paper (Table 1 and 4) did not use different random runs.**
> >
> > **A:** Thanks for your suggestion. We have added the std in Table 1 and Table 4 in the updated version. Each method is tested on 5 random seeds.
> >
> > ----------------------------------
> >
> > **Q7: Societal Impact: No potential negative societal impact.**
> >
> > **A:** Some discussion of societal impact is included in Section 9 L402-L404: “A potential disadvantage of the proposed approach is that biases of the pre-trained LMs may influence its behavior, and further study of LID-based models’ bias is required before they may be deployed in sensitive downstream applications.”
> >
> > The potential negative societal impact caused by the biases of the pre-trained LMs may influence their behavior in sensitive downstream applications. In our experiments, we mitigate this by ensuring that the data used to fine-tune the LMs is free of sensitive content, i.e. the vocabulary of our dataset and the feasible goal space and action space of the VirtualHome environment only describe everyday house chores. We will expand this discussion in any final version of the paper.

---

> > > ### Comment · Reviewer_uuzZ · 2022-08-03
> > > **Reply to Authors' Response**
> > >
> > > Thank you for providing the additional experiments and clarifications. The response addressed my concerns mentioned in the Weaknesses part. I also confirmed the answers to my questions with the updated content of the paper.
> > >
> > > Regarding the Societal Impact part, maybe my previous comment is not clear. My point is that the outcome of this paper may not bring negative social impacts. But I am still happy to see that the authors will give further discussion in the future version.
> > >
> > > Hence, I am glad to increase my rating to 7.

---

### Official Review · Reviewer_5cMv · 2022-07-11

**Rating:** 7
**Confidence:** 3
**Soundness:** 4 excellent
**Presentation:** 4 excellent
**Contribution:** 4 excellent

**Summary:**

This paper presents a framework to use language models like transformers to be used for decision making in interactive environments. The authors also present the features of this approach and present experiments to verify the same. The key contributions are identifying that language models for decision making in RL-type settings improves combinatorial generalization, an active data gathering approach to account for less than ideal pre-collected expert data and how sequentiality is very important in all these encoding techniques for planning in the environments discussed in the paper.

**Questions:**

Q1: In Table 2, VirtualHome seems to be a very tough environment for any policy that isn't LID-ADG. I'd like to see comparison against other methods which have significance performance on this: even though they may not be using ADG, it would be nice to see how much can ADG push a method against a better performing model which might have more expert data.

Q2: Can you provide more intuition/experiments on how different layers of GPT-2 in terms of self-attention are lighting up when a task is being performed? This question is aimed to poke at the structure of the transformer and how different layers are being used for interactive decision making


**Limitations:**

Authors don't discuss the limitations enough of the model in the environment of VirtualHome and BabyAI. I'd like to see a discussion of failure modes of the model and how that traces back to the encoded embeddings.

**Strengths And Weaknesses:**

Strengths: The paper presents the hypotheses very well and also lays out a rigorous enough framework to test the hypotheses. It lays out a framework to use a general large language model to be used in place of existing non-transformer based policy networks. The authors touch on the importance (or lack thereof) of the encoding scheme, how to convert structured pieces of instructions into language and testing the framework for achieving the goal state. Finally the piece about active data gathering is very significant and can be expanded in general to a lot of other RL methods to compensate for lack of expert data in RL environments

Weaknesses: Even though there are experiments in section 7 trying to give an intuitive feel for the reason this method is working, there can be more discussion from a model-structure perspective on how the internal structure of the transformer is creating these embeddings for policy networks.

---

> ### Author Response · Authors · 2022-08-02
> **Response to Reviewer 5cMv (1/2)**
>
> Dear Reviewer, thank you for your valuable comments and feedback. We appreciate that you think our paper is technically solid and the proposed framework is important for RL. We have addressed your questions below. Please let us know if you have any additional questions -- we are happy to clarify or provide additional experiments.
>
> -------------------------------------
>
> **Q1: There can be more discussion from a model-structure perspective on how the internal structure of the transformer is creating these embeddings for policy networks.**
>
> **A:** It is a great suggestion, and we agree that understanding the Transformer architecture is an important research topic. Analyzing how the internal structure of Transformers influences their results is still an active open area of study and is a great direction of study for a future paper.
>
> To provide a preliminary analysis of such structure, in our paper Appendix H, we visualize the attention weights from the self-attention layers of GPT-2. We empirically found that some self-attention layers inside the Transformer assign high attention weights to objects in the goal predicates while some layers focus on the interacted object.
>
> Some recent papers [1,2,3,4] characterize the Transformer architecture from different perspectives. In [1], the authors show that “induction heads play a major role in general in-context learning.” In [4], the authors locate and edit facts stored in language models. Despite these interesting findings, developing a complete understanding of the internal structure of Transformers remains a difficult problem in need of further investigation.
>
> [1] https://transformer-circuits.pub/2022/in-context-learning-and-induction-heads/index.html
>
> [2] https://transformer-circuits.pub/2021/framework/index.html
>
> [3] https://aclanthology.org/2021.emnlp-main.446.pdf
>
> [4] Locating and Editing Factual Associations in GPT
>
> -----------------------------------
>
> **Q2: In Table 2, VirtualHome seems to be a very tough environment for any policy that isn't LID-ADG. I'd like to see comparison against other methods which have significance performance on this: even though they may not be using ADG, it would be nice to see how much can ADG push a method against a better performing model which might have more expert data.**
>
> **A:** The VirtualHome tasks are challenging for RL methods because of the large action spaces, sparse rewards, and long-horizon planning. RL methods perform poorly out of the box, as shown in Table 2, but can be improved by providing additional data / favorable weight initialization, as shown in Table 3.
>
> In Table 3, the offline RL method, Decision Transformer (DT), is trained on data collected by our well-trained model (LID-ADG). Such data can be treated as a type of expert data. When given enough expert data, the offline RL method can work well in solving decision-making tasks in VirtualHome.
>
> In Table 3, the online RL method, PPO, can also solve VirtualHome tasks when its policy is initialized from our well trained model (LID-ADG). This demonstrates that by using a good initialized policy, PPO can solve more challenging tasks.
>
> These experiments show the comparisons of our methods and the stronger baselines that can achieve significant performance in VirtualHome. However, these baselines require extra information during training, such as expert data or weight initialization.

---

> > ### Author Response · Authors · 2022-08-02
> > **Response to Reviewer 5cMv (2/2)**
> >
> > **Q3: Can you provide more intuition/experiments on how different layers of GPT-2 in terms of self-attention are lighting up when a task is being performed?**
> >
> > **A:** Thanks a lot for your suggestion. We have such an experiment in Appendix H, “Visualization of Attention Weights”. In the inference time, when we are decoding the actions, we save the self-attention weights with respect to different layers and different heads. Then, we use BertViz library (https://github.com/jessevig/bertviz) to visualize normalized attention weights as in Figures 11-12. The left side is the query side. The boldness of the lines is proportional to the attention weight.
> >
> > In Figure 11, We show the attention weights of a layer named “Head 3 Layer 2” in dealing with two different tasks. We find that “Head 3 Layer 2” is able to capture objects in the goal predicates, such as “wineglass” and “cutleryfork” in the left figure and “pancake” and “chicken” in the right figure.
> >
> > In Figure 12, we illustrate the attention weights of another two layers named “Head 1 Layer 2” (left) and “Head 4 Layer 738 11” (right). Given the goal predicates, history, and the current observation, the policy predicts the next action as “grab milk”. We find that “Head 1 Layer 2” is able to capture objects in the goal predicates, such as “milk”, “pancake”, and “chicken” while “Head 4 Layer 11” focuses on the interacted object in the predicted action, such as “milk”.
> >
> > We noticed that the attention weights from different self-attention layers are significantly different—some self-attention layers assign high attention weight to objects in the goal predicates while some layers focus on the interacted object. Some layers do not have interpretable meanings.
> >
> > As described in our reply to Q1, we believe it is important to understand the internal structure of Transformers. However, this is still an active open area of study. We would like to explore more in this direction in our future work.
> >
> > -------------------------------------------
> >
> > **Q4: Authors don't discuss the limitations enough of the model in the environment of VirtualHome and BabyAI. I'd like to see a discussion of failure modes of the model.**
> >
> > **A:** Thanks a lot for your suggestion. In the paper we submitted, the limitation of our model is described in L401-404. Some failure cases are shown in Figure 5 and Appendix B.
> >
> > As discussed at L401-404, one drawback of active data gathering is that it relies on hand-designed rules for task relabeling. More generally, a potential disadvantage of the proposed approach is that biases of the pre-trained LMs may influence the model behavior. Further study of LID-based models’ bias is required before they may be deployed in sensitive downstream applications.
> >
> > As shown in Figure 5 in Appendix B, we observed two main types of failure: grounding error and policy error. For failures caused by grounding error, the agent interacts with a wrong object that is not related to the given goal, e.g., the agent puts cutlets instead of the salmon inside the fridge. For failures caused by policy error, the agent cannot find the target objects or does not interact with them.

---

> > > ### Comment · Reviewer_5cMv · 2022-08-05
> > > **Rebuttal done**
> > >
> > > Authors have addressed all my comments and they do have the analysis I was looking for.

---

### Author Response · Authors · 2022-08-02
**General Response to All Reviewers**

We thank the reviewers for their helpful comments and feedback. All the reviewers find our work novel and technically solid. They all agree with the significance of using pre-trained language models (LMs) as a general framework for decision-making tasks. They also believe the active data gathering and the analysis of the encoding scheme in LMs are important. We responded to each of the reviewers’ questions below. We also updated the draft based on the reviewers’ comments (changes are highlighted) and attached the code for training LID-Text in the supplementary material.

---

### Meta-Review · Area_Chair_fAno · 2022-08-25

**Recommendation:** Accept
**Confidence:** Certain

**Metareview:**

This paper adapts the "pretrain-then-finetune" framework to policy learning using large language models and demonstrates its effectiveness. They also develops an active expert data gathering approach for settings where no expert data is available. All reviewers find the empirical findings in the paper interesting and the work technically solid. This paper may spur more work in using pretrained language models in RL settings. I recommend acceptance.

**Award:**

No

---

### Decision · Program_Chairs · 2022-09-14

Accept